# Monitoring of Nonadiabatic Effects in Individual Chromophores by Femtosecond Double-Pump Single-Molecule Spectroscopy: A Model Study

**DOI:** 10.3390/molecules24020231

**Published:** 2019-01-09

**Authors:** Maxim F. Gelin, Elisa Palacino-González, Lipeng Chen, Wolfgang Domcke

**Affiliations:** Department of Chemistry, Technische Universität München, D-85747 Garching, Germany; elisa.palacino@ch.tum.de (E.P.-G.); chen0846@gmail.com (L.C.); domcke@ch.tum.de (W.D.)

**Keywords:** single-molecule spectroscopy, nonadiabatic dynamics, weak-field regime, strong-field regime

## Abstract

We explore, by theoretical modeling and computer simulations, how nonadiabatic couplings of excited electronic states of a polyatomic chromophore manifest themselves in single-molecule signals on femtosecond timescales. The chromophore is modeled as a system with three electronic states (the ground state and two non-adiabatically coupled excited states) and a Condon-active vibrational mode which, in turn, is coupled to a harmonic oscillator heat bath. For this system, we simulate double-pump single-molecule signals with fluorescence detection for different system-field interaction strengths, from the weak-coupling regime to the strong-coupling regime. While the signals are determined by the coherence of the electronic density matrix in the weak-coupling regime, they are determined by the populations of the electronic density matrix in the strong-coupling regime. As a consequence, the signals in the strong coupling regime allow the monitoring of nonadiabatic electronic population dynamics and are robust with respect to temporal inhomogeneity of the optical gap, while signals in the weak-coupling regime are sensitive to fluctuations of the optical gap and do not contain information on the electronic population dynamics.

## 1. Introduction

Starting from the mid 1980s, the monitoring of vibrational wave packets and the making/breaking of chemical bonds with femtosecond time resolution has been explored for molecular ensembles [1]. Recently, molecular spectroscopists made the next significant step by looking at the dynamics of individual molecules on femtosecond timescales. Femtosecond resolution was brought to the single-molecule (SM) spectroscopy community by van Hulst and coworkers [2,3]. Their technique combines fluorescence detection of SMs pioneered by Orrit and Bernard [4] with pulse shaping and phase-locked double-pump excitation, pioneered in femtosecond ensemble molecular spectroscopy by Scherer and co-workers [5]. The scanning of SM fluorescence vs time delay between the pulses combines fluorescence detection (which is usually associated with nanosecond time scale) with femtosecond time resolution.

The double-pump SM experiments of Refs. [6,7,8,9,10] were performed with highly photostable chromophores which can be adequately described by a model with a single excited electronic state and a single Condon-active vibrational mode. Oscillatory transients detected in these experiments deliver information on distributions of vibrational frequencies and electronic dephasing times of different chromophores and reveal, predominantly, heterogeneity of the ensemble of chromophores embedded in a polymer matrix at ambient temperature [6,7,8,9,10,11,12,13]. On the other hand, double-pump SM experiments performed on LH2 antenna complexes of purple bacteria [14] and their theoretical analysis [15,16,17] revealed important information on electronic interstate couplings and photophysical processes in light-harvesting complexes which cannot be obtained in ensemble spectroscopy.

It is thus of interest to explore more systematically to what extent information on the dynamics of coupled electronic states in individual chromophores can be extracted from double-pump SM signals with fluorescence detection. One of the key intramolecular processes is radiationless decay, which is caused by nonadiabatic coupling of electronic states of polyatomic chromophores [18]. As established by nonlinear femtosecond ensemble spectroscopy, these couplings are reflected by oscillatory signals in the time domain which may have electronic [19,20] or vibrational [21] character. On the other hand, nonadiabatic couplings cause efficient and fast depopulation of excited electronic states, increasing thereby chromophore’s photostabilty. In the present work, we depart from the description of a chromophore as an electronic two-state system and study, by computer simulations of simple models, how nonadiabatic coupling among excited electronic states is manifested on the femtosecond timescale in double-pump SM signals.

## 2. Theoretical Framework

### 2.1. Hamiltonian, Master Equation and the SM Signal

Let us consider a chromophore embedded in a polymer matrix. As revealed by femtosecond nonlinear ensemble spectroscopy [21], a single effective high-frequency (reaction) mode may dominate responses of molecular systems at short time scales. We thus include three electronic states coupled to a single Condon-active vibrational mode of the chromophore in the system. The remaining vibrational modes of the chromophore as well as vibrational modes of the polymer matrix are treated as a thermal environment. The effects of the environment are accounted for by an appropriate master equation. A method for the microscopic construction of such reduced dimensionality models and parametrization of the model Hamiltonians can be found, e.g., in Refs. [22,23,24].

In the diabatic representation, the system Hamiltonian has the form
(1)$$H_{S}=\sum_{k=0,1,2}|e_{k}\rangle \left(h_{k}+\epsilon_{k}\right)\langle e_{k}|+v\left(|e_{1}\rangle \langle e_{2}\left|+\right|e_{2}\rangle \langle e_{1}|\right).$$

Here $|e_{0}\rangle$ is the electronic ground state, $|e_{1}\rangle$ and $|e_{2}\rangle$ are two excited electronic states (higher lying electronic states can straightforwardly be included, if necessary), $\epsilon_{1}$ and $\epsilon_{2}$ are the electronic excitation energies ($\epsilon_{0}=0$) and *v* is the electronic coupling of the states $|e_{1}\rangle$ and $|e_{2}\rangle$. The vibrational Hamiltonians are assumed to be harmonic,
(2)$$h_{k}=\frac{\Omega }{2}\left\{P^{2}+{\left(Q-Q_{k}^{\left(0\right)}\right)}^{2}\right\},$$
k=0,1,2. Here *Q* and *P* are the dimensionless coordinate and momentum of the vibrational mode, Ω is its frequency, and $Q_{k}^{\left(0\right)}$ are the horizontal shifts of the potential energy functions with respect to the electronic ground state ($Q_{g}^{\left(0\right)}=0$). The chromophore interacts with a pair of phase-locked pulses as specified by the system-field Hamiltonian
(3)$$H_{F}\left(t\right)=-\left[E\left(t\right)X^{†}+E^{*}\left(t\right)X\right].$$

Here
(4)$$\begin{array}{ccc} E(t) & = & E_{1}\left(t\right)+E_{2}\left(t\right),\\ E_{1}\left(t\right) & = & E_{0}f\left(t+\tau \right)e^{-i\omega_{1}t},\\ E_{2}\left(t\right) & = & E_{0}f\left(t\right)e^{i(\phi -\omega_{2}t)}, \end{array}$$
and
(5)$$X=\sum_{k=1,2}\hat{e}\mu_{0k}|e_{0}\rangle \langle e_{k}|,\;\;\;X^{†}=\sum_{k=1,2}\hat{e}\mu_{0k}|e_{k}\rangle \langle e_{0}|,$$
are the transition dipole operators, $\mu_{01}$ and $\mu_{02}$ are the electronic transition dipole vectors, and $\hat{e}$ is the unit vector of the polarization of the two pump pulses. Equation (Equation 4) describes pump pulses with the amplitude $E_{0}$, dimensionless envelope f(t) and carrier frequencies $\omega_{1}$ and $\omega_{2}$; τ is the time delay between the pulses and ϕ is their relative phase. We set the arrival time of the first pulse at t=−τ, while the second pulse arrives at t=0. Following [11,12], it is convenient to introduce the system-field coupling parameters
(6)$$\eta_{0k}=E_{0}\left(\hat{e}\mu_{0k}\right),\;\;\;k=1,\;2.$$

To account for environment-induced relaxation and homogeneous dephasing, we adopt the system-bath approach and write the total Hamiltonian as the sum of the system Hamiltonian, the Hamiltonian of the heat bath, and their coupling,
(7)$$H=H_{S}+H_{B}+H_{SB}.$$

We further assume that the system is bilinearly coupled through its mode *Q* to a harmonic bath:(8)$$H_{SB}=Q\sum_{a}c_{a}q_{a},$$
(9)$$H_{B}=\frac{1}{2}\sum_{a}\omega_{a}\left\{p_{a}^{2}+q_{a}^{2}\right\}.$$

Here $p_{a}$, $q_{a}$ and $\omega_{a}$ are the dimensionless momentum, coordinate, and frequency of *a*th oscillator of the bath, and the system-bath coupling constants $c_{a}$, for simplicity, are taken the same for all electronic states involved. The influence of the bath on the system dynamics is determined by the spectral density
(10)$$g\left(\omega \right)=\sum_{a}c_{a}^{2}\delta \left(\omega -\omega_{a}\right).$$

The main photophysical processes in the chromophore are governed by the system Hamiltonian $H_{S}$ and will be treated numerically exactly. We assume that the remaining modes of the chromophore and the polymer matrix are weakly coupled to the system and can be treated perturbatively. Retaining the terms up to the second order in $H_{SB}$ and assuming that the bath is fast on the system dynamics timescale (so-called Markovian approximation) one can derive a master equation for the reduced density operator ρ(t) of the system,
(11)$$\frac{\partial }{\partial t}\rho \left(t\right)=-\frac{i}{\hbar }\left[H_{S}+H_{F}\left(t\right),\rho \left(t\right)\right]+\left(R+D\right)\rho \left(t\right),$$
where R is the multilevel Redfield relaxation operator (see Refs. [22,23,24,25,26] for detailed derivations). The operator
(12)$$D\rho \left(t\right)=-\gamma \sum_{k=1,2}|e_{0}\rangle \langle e_{k}|\langle e_{0}\left|\rho \left(t\right)\right|e_{k}\rangle +H.c.$$
describes pure electronic dephasing, where γ is a phenomenological dephasing rate. A microscopic treatment of electronic dephasing is also possible (see, e.g., Ref. [27]), but the phenomenological description via Equation (Equation 12) is sufficient for the purposes of the present work [8,11,28].

The double-pump SM signal is defined as the total (time- and frequency-integrated) fluorescence of a single chromophore detected as a function of the interpulse delay τ. This signal is proportional to the time integral of the population of the excited electronic states of the chromophore and can be evaluated through the reduced density matrix as [29]
(13)$$S\left(\tau \right)∼\int_{t_{0}}^{\infty }dt\mathrm{Tr}\left\{X\rho \left(t\right)X^{†}\right\},$$
where $t_{0}$ is any time moment before the arrival of the first pump pulse such that $\mathrm{Tr}\{X\rho \left(t_{0}\right)X^{†}\}=0$. Note that an alternative definition of the signal, $S\left(\tau \right)∼\mathrm{Tr}\{X\rho \left(t_{f}\right)X^{†}\}$ ($t_{f}$ being a time moment at which the second pump pulse is over), which has been used in the simulations of Refs. [11,12], is not applicable in the present case owing to the presence of the electronic coupling *v*.

The embedding in a polymer matrix leads to a highly heterogeneous ensemble of chromophores. Each chromophore experiences thermal fluctuations and the values of the parameters
(14)$$\epsilon_{k},\;\;Q_{k}^{\left(0\right)},\;\;\eta_{0k},\;\left(k=1,\;2\right)\;\;\mathrm{as}\;\mathrm{well}\;\mathrm{as}\;\;\Omega ,\;\;v,\;\;\xi ,\;\;\gamma$$
may be different at each moment
(15)$$\tau_{j}=j\Delta_{\tau },\;\;j=0,1,2,...$$
of the detection of the SM signal $S(\tau_{j})$ (e.g., $\Delta_{\tau }=3$ fs in SM experiments of Refs. [6,7,8] and $\Delta_{\tau }=25$ fs in Ref. [14]). It should be stressed that the time interval between the measurements corresponding to different $\tau_{j}$ is much longer than any relevant microscopic time interval specifying electron-vibrational dynamics and fluorescence detection of the individual chromophore. Hence, there is no correlation between the values of parameters (Equation 14) in any two consecutive measurements. To simulate this measurement protocol, we introduce a stochastic modulation of the chromophore parameters [11,12],
(16)$$A_{\tau }=\bar{A}+\delta_{A}\left(r_{\tau }-1/2\right),$$
were $A_{\tau }$ is a stochastic realization of any parameter from the list (Equation 14) at a specific time delay τ, $\bar{A}$ is its mean value, $\delta_{A}$ is the amplitude of modulations, and $r_{\tau }$ is a random number which is uniformly distributed in the interval [0,1].

The intensity of the double-pump SM signal can be expanded in the system-field coupling as [11,12,27]
(17)$$S\left(\tau \right)=\sum_{k=2,4,6,...}S_{k}\left(\tau \right),$$
where *k* corresponds to the number of interactions of the chromophore with the laser pulses and $S_{k}\left(\tau \right)∼E_{0}^{k}$. For sufficiently weak pulses, the signal is represented by $S_{2}\left(\tau \right)$, scales linearly with the pulse intensity, and can be expressed through the linear response function [11]. For stronger chromophore-field coupling, higher-order terms in the expansion (Equation 17) are relevant. The *k*th contribution can be decomposed in two terms, $S_{k}\left(\tau \right)=S_{k}+{\tilde{S}}_{k}\left(\tau \right)$. Here $S_{k}$ describes a τ-independent background, which results from the interaction of the chromophore with just one of the pump pulses. The τ-dependent contribution ${\tilde{S}}_{k}\left(\tau \right)$ stems from the interaction of the chromophore with both pulses. Hence the total signal can be represented as the sum of a constant background and a τ-dependent part,
(18)$$S\left(\tau \right)=S_{\infty }+\tilde{S}\left(\tau \right)$$
where
$$S_{\infty }=\sum_{k}S_{k},\;\;\;\tilde{S}\left(\tau \right)=\sum_{k}{\tilde{S}}_{k}\left(\tau \right).$$

As has been shown in Ref. [12], the SM signal can be conveniently separated into population and coherence contributions. The separation remains valid in the present case, but the formulas of Ref. [12] have to be slightly modified: If the pump pulses are temporally well separated (that is, if the time interval between the pulses, τ, is much longer than the pulse duration $\tau_{p}$), the double-pump signal can be approximated as
(19)$$S\left(\tau \right)=A\left(\tau \right)+\left(B\left(\tau \right)e^{i\phi }+B^{*}\left(\tau \right)e^{-i\phi }\right)e^{-(\gamma +\gamma_{\xi })\tau }.$$

Here γ is the electronic dephasing rate, while $\gamma_{\xi }$ is an extra rate induced by the nonadiabatic intrastate coupling *v* as well as by the coupling of the chromophore to the heat bath. A(τ) is the contribution which results from the evolution of the chromophore in the electronic population in the states $|e_{0}\rangle \langle e_{0}|$ and $|e_{2}\rangle \langle e_{2}|$. B(τ) is the coherence contribution which involves the coherences $|e_{0}\rangle \langle e_{2}|$ and $|e_{2}\rangle \langle e_{0}|$. In the present work, we do not try to derive Equation (Equation 19) analytically, but rather use it as a convenient tool for the interpretation and discussion of SM signals.

### 2.2. Computational Details

The mean values of the model parameters (1) are designated by an overbar and are selected as follows. The vibrational frequency is set to $\bar{\Omega }=0.151$ eV, which yields a vibrational period $\tau_{\bar{\Omega }}=2\pi /\bar{\Omega }=27$ fs. The dimensionless shifts of the excited-state potential energy functions are fixed at ${\bar{Q}}_{2}^{\left(0\right)}=0.3$ and ${\bar{Q}}_{1}^{\left(0\right)}=-1.7$, the electronic energy difference ${\bar{\epsilon }}_{2}-{\bar{\epsilon }}_{1}=0.74$ eV, and the intra-state coupling is v=0.05 eV. The potential energy functions of the states $|e_{1}\rangle$ and $|e_{2}\rangle$ cross at Q=1.8 (see Figure 1). The chromophore parameters are taken from a recent model describing the photophysics of the *B* and $Q_{y}$ states of free-base tetraphenylporphyrin [30]. Obviously, SM signals depend on specific values of the model parameters. However, as we argue below, the qualitative behavior and interpretation of the signals of chromophores with nonadiabatic couplings is generic and model independent.

We assume that the state $|e_{2}\rangle$ is optically bright from the ground state $|e_{0}\rangle$, while the state $|e_{1}\rangle$ is optically dark, $\mu_{01}=0$ (this is common for polyatomic chromophores with coupled excited electronic states). In this case, the SM signal can be evaluated by a simplified version of Equation (Equation 13),
(20)$$S\left(\tau \right)∼\int_{t_{0}}^{\infty }dt\mathrm{Tr}\{\langle e_{2}\left|\rho \left(t\right)\right|e_{2}\rangle \},$$
and a single parameter $\eta_{02}$, which will be varied, determines the coupling of the chromophore with the external field of the pump pulses. The latter have Gaussian envelopes,
(21)$$f\left(t\right)=\exp\{-{\left(t/\tau_{p}\right)}^{2}\}$$
($\tau_{p}=10$ fs being the pulse duration, $\tau_{p}\ll \tau_{\bar{\Omega }}$) and identical carrier frequencies ($\omega_{1}=\omega_{2}$). The detuning frequency
(22)$${\bar{\omega }}_{det}=\omega_{1}-{\bar{\epsilon }}_{2}/\hbar$$
is fixed at 0.107 eV, which corresponds to an excitation between the first and second vibrational levels of the bright state $|e_{2}\rangle$. The relative phase of the two pump pulses is set to zero (ϕ=0). General properties of ϕ-dependence of SM signals have been discussed in Refs. [11,12].

The vibrational relaxation operator R in the master Equation (Equation 11) is described by multi-level Redfield theory [22,23,24,25,26] with an Ohmic spectral density,
(23)g(ω)=ξωexp{−ω/Ω},
where ξ is a dimensionless parameter which controls the rate of vibrational energy redistribution in the chromophore. The explicit dependence of R on the external fields can be neglected for the pulses employed in the present work (see discussion in Ref. [31]). The electronic dephasing rate is chosen as γ=0.01 eV ($\gamma^{-1}=66$ fs), which is typical for the experiments of Refs. [6,7]. The temperature is set to T=300 K. In this case, $\coth[\hbar \bar{\Omega }/\left(2k_{B}T\right)]\approx 1$ and the chromophore resides initially in its ground vibrational state in the electronic ground state,
(24)$$\rho \left(t_{0}\right)=\left|0\rangle \langle 0|\right|e_{0}\rangle \langle e_{0}|.$$

Static disorder in the electronic energy gap is usually the main source of inhomogeneous broadening in ensemble experiments, and a typical amplitude of the electronic energy modulations is of the order of several 100 cm${}^{-1}$ at ambient temperatures [32,33]. In the present work, we set $\delta_{\epsilon }=150$ cm${}^{-1}$ (0.0186 eV). As has been established in Refs. [11,12], disorder in other parameters produces qualitatively similar changes in SM signals and is not considered in the present work.

The procedure of the calculation of the SM signal is briefly described as follows. According to Equation (Equation 16), we generate a realization of $r_{\tau }$ for each time delay τ and calculate the snapshot electronic energies $\epsilon_{1}={\bar{\epsilon }}_{1}+\delta_{\epsilon }\left(r_{\tau }-1/2\right)$ and $\epsilon_{2}={\bar{\epsilon }}_{2}+\delta_{\epsilon }\left(r_{\tau }-1/2\right)$. With these values of $\epsilon_{1}$ and $\epsilon_{2}$, the driven snapshot master Equation (Equation 11) is converted into matrix form by an expansion in terms of the eigenstates of the system Hamiltonian $H_{S}$ (which becomes a 45×45 matrix) and solved via the fourth-order Runge-Kutta integrator with a time step 0.5 fs. The so obtained ρ(t) is used for the numerical evaluation of S(τ) via Equation (Equation 20).

## 3. SM Signals

According to Equations (Equation 10) and (Equation 23), the parameter ξ controls the coupling of the chromophore to intra- and inter-molecular vibrational modes and is proportional to the total Huang-Rhys factor of these modes. Since SM signals depend sensitively on ξ, we consider two representative cases: ξ=0.014 (model I, weak coupling to the bath) and ξ=0.042 (model II, intermediate coupling to the bath). If v=0, the signals become ξ-independent and models I and II reduce to the shifted harmonic oscillator model considered in Refs. [11,12].

A comprehensive picture of the dependence of the SM signals S(τ) on the system-field coupling $\eta_{02}$ in models I and II is given by Figure 2 and Figure 3, respectively. We start from a brief overview of the signals. In all figures, black lines give a reference picture showing the signals calculated without parameter modulations. Blue lines depict the signals calculated with stochastic modulations of the electronic energy gap as explained in Section 2.2. The intensity of the signals is given in arbitrary units, since only relative intensities are meaningful. According to Equation (Equation 18), all signals S(τ) reveal a constant background $S_{\infty }$ and a τ-dependent $\tilde{S}\left(\tau \right)$ contribution.

Figure 2 shows SM signals S(τ) for model I. Panel (a) depicts the signal in the limit of weak system-field coupling. The signal exhibits damped oscillations which, due to a relatively small shift of the potential energy function of the bright state with respect to the ground state, reveal the detuning frequency with a period $2\pi /{\bar{\omega }}_{det}=39$ fs (see the discussion in Ref. [11]). The damping is largely caused by electronic dephasing ($\gamma^{-1}=66$ fs). After τ>250 fs, $S\left(\tau \right)\approx S_{\infty }$ and does not contain any dynamic information. The signal is qualitatively described by Equation (Equation 19), in which the coherence contribution B(τ) is responsible for τ-dependent evolution ($\gamma \gg \gamma_{\xi }$), while the population contribution A(τ) is τ-independent and is responsible for the constant background $S_{\infty }$. Hence, the nonadiabatic dynamics is not manifested in the SM signal in panel (a), which looks qualitatively like the signal of a displaced harmonic oscillator in the limit of weak system-field coupling [11].

Panel (b) corresponding to $\eta_{02}=0.05$ eV reveals a turnover from the weak-coupling regime to the strong-coupling regime. For a large transition dipole moment of 1 atomic unit (2.54 D), this value of the system-field coupling can be recalculated into the power density $∼10^{11}$ W/cm${}^{2}$, which is a relatively moderate intensity. For v=0, the border line between the weak-coupling regime and the strong-coupling regime corresponds to a somewhat smaller value of $\eta_{02}\approx 0.03$ eV [12,34]. In comparison with the signals in panel (a) the signals in panel (b) start to undergo qualitative changes. Namely, the black line ($\delta_{\epsilon }=0$) in panel (b) shows a low-amplitude beating with a period $\tau_{\bar{\Omega }}=27$ fs for τ>250 fs and a small hump starts do develop around τ≈300 fs. If the electronic energy modulations are taken into account, these new features are buried in the noise (blue line).

If the system-field coupling becomes stronger (panels c and d), the features emerging in panel (b) become much more pronounced. The amplitude of vibrational beatings a period $\tau_{\bar{\Omega }}=27$ fs in S(τ) increases and the hump around τ≈300 fs develops into a well visible maximum (black lines). It should be noted that the noise of the energy-gap fluctuations is suppressed with increasing system-field coupling.

The hump around τ≈300 fs in Figure 2c,d is the signature of the nonadiabatic coupling between the bright and dark electronic states. It represents a recurrence of the population of the bright electronic state which is driven by coherent vibrational motion of the Condon active mode. Such electronic recurrences, which are typical for a large variety of nonadiabatic systems [26,31], are monitored by time- and frequency-resolved fluorescence signals [35,36].

The reasons for significant amplification of the vibrational and vibronic features and the robustness of the signals with respect to disorder in the electronic energy gap can be explained by inspection of Equation (Equation 19). For $\tau >\gamma^{-1}$, the SM signal is represented by the population contribution A(τ). In the strong-coupling limit (k≥4 in Equation (Equation 17)), this contribution reveals vibrational wavepacket motion in the electronic ground state $|e_{0}\rangle$ and vibronic wavepacket motion in the coupled excited states $|e_{1}\rangle$ and $|e_{2}\rangle$. A(τ) is unaffected by electronic dephasing, but is governed by the combined effects of vibrational relaxation (ξ) and electronic coupling (*v*). As can be seen in Figure 2c,d, τ∼100 fs corresponds to turnover from the regime dominated by the coherence contribution B(τ) (which reveals the detuning frequency ${\bar{\omega }}_{det}$) to the regime dominated by the population contribution A(τ) (which reveals the vibrational frequency $\bar{\Omega }$ and the electronic recurrence). The robustness of strong-field SM signals to static disorder can also be explained in terms of Equation (Equation 19): The coherence contribution B(τ) is much more sensitive to modulations of the electronic energy than the population contribution A(τ).

To follow more closely changes in the oscillatory features of S(τ) with the system-field coupling, it is convenient to introduce the time moments $\tau_{n}^{max}$ corresponding to the local maxima $S(\tau_{n}^{max})$ of the SM signal. Figure 4 shows the distances between the local maxima, $\Delta_{n}^{max}=\tau_{n+1}^{max}-\tau_{n}^{max}$ vs *n* for the signals of Figure 2a–d. $\Delta_{n}^{max}$ corresponding to Figure 2a (blue stars) reveal exclusively oscillations with the detuning frequency, $2\pi /{\bar{\omega }}_{det}=39$ fs. As the system-field coupling increases, $\Delta_{n}^{max}$ as a function of *n* can be decomposed into three domains. Domain 1 reveals oscillations with the detuning frequency, domain 2 is a transient region, and domain 3 reveals purely vibrational oscillations with a period of $\tau_{\bar{\Omega }}=27$ fs. As the system-field coupling increases (cf. circles, diamonds and triangles in Figure 4), domains 1 and 2 shrink, while domain 3 extends owing to the increasing role of the population contribution to the SM signal.

Let us now consider the SM signals for model II (Figure 3), which corresponds to a stronger system-bath coupling and hence faster vibrational energy relaxation. The signal in the weak-field regime is shown in panel (a). It looks similar to the corresponding signal for model I (Figure 2a). It reveals oscillations with the detuning frequency ${\bar{\omega }}_{det}$, but decays somewhat faster (at τ>200 fs, $S\left(\tau \right)\approx S_{\infty }$). In terms of Equation (Equation 19), the τ-dependence of the signal is given by the coherence contribution B(τ), which decays due to electronic dephasing γ (which is responsible for ≈80% of the decay rate) and due to the coupling to the bath $\gamma_{\xi }$ (which is responsible for ≈20% of the decay rate). The population contribution A(τ) produces the constant background $S_{\infty }$. The nonadiabatic coupling is reflected by an additional decay which, however, does not change the qualitative behavior of S(τ). One can conclude that SM signals in the weak-field regime do not exhibit signatures of nonadiabatic dynamics.

Figure 3b depicts the signal corresponding to turnover from the weak system-field coupling regime to the strong system-field coupling regime. The black line ($\delta_{\epsilon }=0$) shows low-amplitude beatings with a period $\tau_{\bar{\Omega }}=27$ fs for τ>200 fs, which can hardly be distinguished when electronic energy modulations are taken into account (blue line). When the system-field coupling is further increased (Figure 3c,d), vibrational beatings gain amplitude and the effect of the energy gap modulation decreases, since the signal is dominated by the population contribution A(τ). The signals in Figure 3b–d have similar intensities, but the value of $S_{\infty }$ (which determines position the baseline in the figures) depends strongly on the system-field coupling $\eta_{02}$. This is the signature of the regime of strong system-field coupling, in which the signal intensity is controlled by the Rabi frequency ${\bar{\Omega }}_{R}$ and exhibits an oscillatory dependence ($S_{\infty }∼\sin^{2}\left({\bar{\Omega }}_{R}\tau_{p}/2\right)$) on $\eta_{02}$ [12,34].

Owing to relatively strong system-bath coupling, the electronic recurrence around τ≈300 fs is weak in model II. However, the signals in Figure 3c,d show remnants of this feature: the overall increase of the signal from 150<τ<300 fs. This increase cannot be caused by any relaxation process and is a clear signature of nonadiabatic coupling. This is reminiscent of the behavior of the SM signal that was reported but remained unexplained in Ref. [7] (see Figure 3 and the pertinent discussion).

## 4. Conclusions

To explore the feasibility of monitoring ultrafast nonadiabatic dynamics via femtosecond double-pump SM spectroscopy, we performed a series of simulations of SM signals of a chromophore possessing a pair of coupled excited electronic states. Our results can be briefly summarized as follows. The signals in the weak-field regime (which scale linearly with the intensity of the pump pulses) do not reveal information on the nonadiabatic dynamics. In this regime, the signals of chromophores with and without electronic interstate couplings are qualitatively indistinguishable. The signals in the strong-field regime (which is characterized by nonlinear scaling of the signal with the pulse pump-pulse intensity) allow the monitoring of the nonadiabatic population transfer in real time. The electronic recurrences in the signals are the signatures of the nonadiabatic dynamics.

The weak-field/strong-field regimes in SM spectroscopy are not only governed by the strength of the laser pulses, but also by the orientation of the selected chromophore (due to the scalar product of the excitation dipole and the polarization vector of the incident electric field). Although the possibility of the detection of double-pump SM signals in the strong-field regime was demonstrated [6], the SM signals detected for different chromophores may correspond either to the weak-coupling regime or to the strong-coupling regime, because different chromophores have different orientations in a polymer matrix [2]. The present work demonstrates that the SM signals of chromophores with coupled electronic states are qualitatively different in the two regimes, and the information content of femtosecond double-pump single-molecule signals is enhanced with the system-field coupling strength.

In the weak-field regime, double-pulse SM spectroscopy is a linear technique. It monitors the evolution of electronic coherence of the density matrix of the chromophore which is not sensitive to electronic population transfer and decays on the timescale of electronic dephasing. In the strong-field regime, on the other hand, the signals reveal the evolution of electronic populations of the density matrix of the chromophore which are not affected by electronic dephasing and are robust with respect to energy gap fluctuations. As is well known, third-order transient-absorption pump-probe (and other more sophisticated 4-wave-mixing techniques) of femtosecond ensemble spectroscopy to monitor the dynamics of electronic populations [37]. In this context, a recent extension of femtosecond SM spectroscopy towards detection of transient absorption of individual chromophores (triple-pulse signals) [38] looks promising.

## Figures and Tables

**Figure 1 molecules-24-00231-f001:**
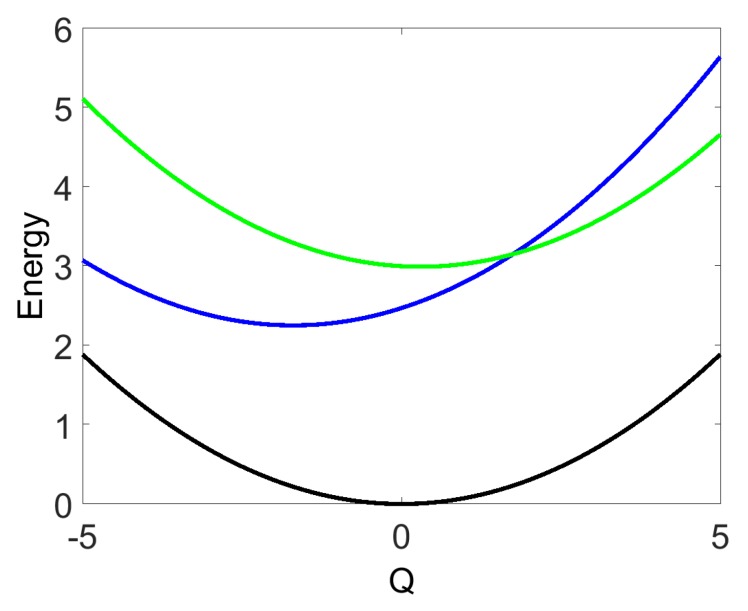
Sketch of the potential-energy functions of the chromophore with the electronic ground state $|e_{0}\rangle$ (black) and coupled excited electronic states $|e_{1}\rangle$ (blue) and $|e_{2}\rangle$ (green).

**Figure 2 molecules-24-00231-f002:**
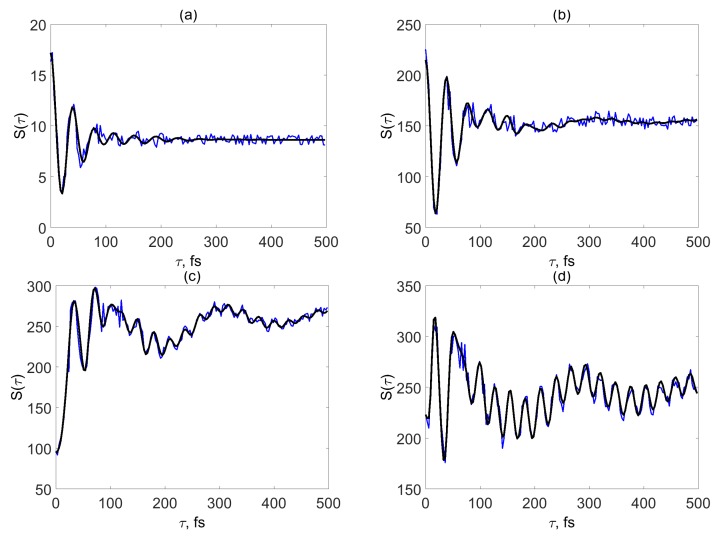
Model I. SM signals for different system-field couplings. $\eta_{02}=$ 0.01 eV (**a**), 0.05 eV (**b**), 0.11 eV (**c**), and 0.17 eV (**d**). Black lines show the single molecule (SM) signals calculated without parameter modulations. Blue lines depict the SM calculated with modulations of electronic energies.

**Figure 3 molecules-24-00231-f003:**
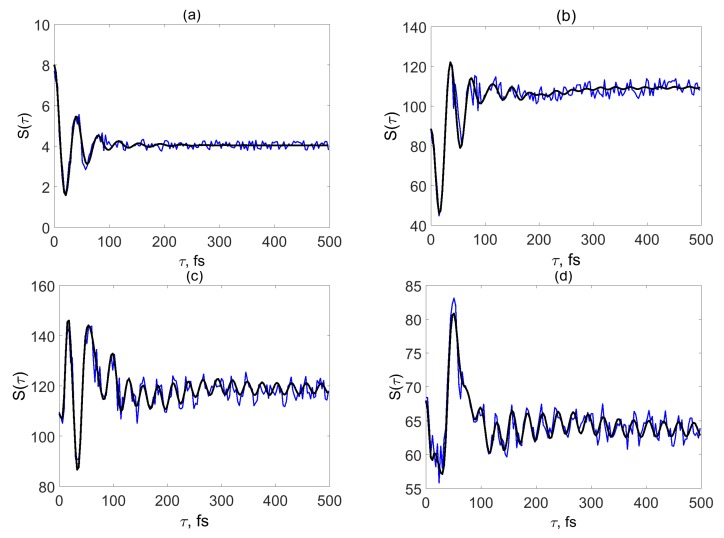
Model II. SM signals for different system-field couplings. $\eta_{02}=$ 0.01 eV (**a**), 0.09 eV (**b**), 0.17 eV (**c**) and 0.19 eV (**d**). Black lines show the SM signals calculated without parameter modulations. Blue lines depict the SM calculated with modulations of electronic energies.

**Figure 4 molecules-24-00231-f004:**
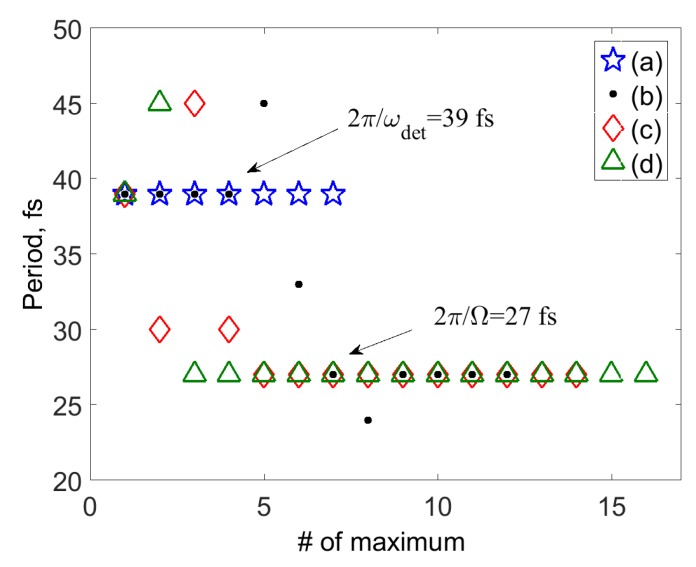
Model I. Distances $\Delta_{n}^{max}=\tau_{n+1}^{max}-\tau_{n}^{max}$ between the adjacent local maxima $S(\tau_{n}^{max})$ of the SM signals vs n=1,2,.... Different symbols correspond to panels (**a**–**d**) of Figure 2 as indicated in the legend.

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
