# Peer review of "Monitoring of Nonadiabatic Effects in Individual Chromophores by Femtosecond Double-Pump Single-Molecule Spectroscopy: A Model Study"

_molecules, 2019, doi:10.3390/molecules24020231_

Round 1
Reviewer 1 Report
Find attached.

Author Response
1. In an actual experiment, higher excitation flux can lead to sample photobleaching and other unwanted nonlinear effects. How do the authors propose to overcome this issue in the predicted method? Have they calculated the safe range of power for samples such as light harvesting proteins?
Answer. The chromophores suitable for single-molecule spectroscopy have to be judicially selected to exhibit exceptional photostability. Moreover, the range of the pulse intensities has to be carefully tuned to avoid destruction of the chromophore. These issues are well known in the field of single-molecule (SM) spectroscopy and were discussed in detail, for example, in previous work (Ref 12 of the manuscript). Here we wish to emphasize the following aspects.
(i) Strong-field effects in double-pump SM spectroscopy have been experimentally observed and exploited. An earlier variant of SM pump-probe experiments by van Hulst and coworkers relied on strong lasers pulses and saturation effects [van Dijk et al, J. Chem. Phys. 2005, 123, 064703]. The same experimental method has also been successfully applied to light harvesting complexes LH2 of purple bacteria [P. Maly et al, Proc. Nat. Acad. Sci., 2016, 113, 2934]. In these experiments, the phases of the laser pulses were not controlled and the time resolution was limited to several hundreds of femtoseconds. More recent femtosecond double-pump SM experiments with phase-locked pulses were performed with weak-to-moderate pump pulses, although the possibility of detection of SM signals in the strong-field regime (with population inversion) was clearly demonstrated [Ref 6 of the present manuscript]. As has been pointed out by van Hulst and coworkers, weak-field and strong-field regimes ”are two extremes on a continuum of increasing intensity. Our experiments morph quite naturally from weak-field into strong field depending on the orientation of the molecule in the sample, i.e. the overlap between the excitation dipole and the incident electric field “ [Ref 2 of the present manuscript].
(ii) The turnover from the weak-coupling regime to the strong-coupling regime corresponds to a coupling parameter eta_02 = 0.05 eV. For a large transition dipole moment of 1 atomic unit (2.54 D), this corresponds to a power density of ~ 10^11 W/cm^2, which is a moderate intensity.
Changes. We included (a slightly modified version of) (ii) in Sec. III.
2. Please provide a Fourier transform of the time traces to show the change in oscillation frequency as a function of system field coupling. How many electronic recurrences do the authors observe? A longer time trace will be helpful in this matter.
Answer. Due to the nonadiabatic coupling of the electronic states of the chromophore (resulting in population transfer from the bright electronic state to the dark electronic state) as well as due to the coupling of the chromophore to the environment, the amplitude of the useful (oscillatory) signal decreases rapidly with time. This is why just a single electronic coherence is visible in the signal, and performing the Fourier transform does not give additional insight. Instead, we added a new figure (Fig 3 of the revised manuscript), in which we display and analyze the frequency changes in a more transparent manner.
3. How do the signal for the strong field interaction differ from the ones shown in Ref. 12?
Answer. In Ref. 12, nonadiabatic coupling of electronic states was absent. There are two main qualitative differences due to electronic inter-state coupling. First, the electronic inter-state coupling produces a hump (electronic population recurrence) in the SM signal. Second, the signals of Ref 12 were independent of the coupling to the environment, while the signals of the present work depend sensitively on the system-bath coupling. We emphasize these effects in the revised manuscript.
Reviewer 2 Report
The authors present a fundamental study about the potential evidences of non-adiabatic coupling in single-molecules (SM) spectroscopy, by adopting a model hamiltonian that includes a coupling with an external bath and it is dynamically coupled to external radiating field.
By exploring different regimes of coupling, the authors highlight when non-adiabatic coupling between electronic states manifests itself.
The text is enough clear, and the physics of the presented results is sounding, although the impact on interpretation of experiments is only vaguely discussed and should be probably emphasized.
The introduction of stocastic modulation of electronic parameters sounds a bit critical in the current form, since: (a) the modulation of each parameter is completely uncorrelated one from each others, (b) the amplitude of the stochastic modulation double the numbers of parameters, (c) the added physics is negligible (Fig. 2 and 3).
About the parametrization and the choice of the model hamiltonian, the authors might find interesting some other work in the literature, quite relevant even if applied to a different phenomenon (see for instance Di Maiolo et al. [J. Chem. Theory Comput., 2018, 14 (10), pp 5339–5349], and other works by the same group).
I found somehow confusing the usage of different scale for intensity in Figures 2(a-d) and 3(a-d), making less easy to read the extent of the fluctuations and the comparisons. The authors could consider to add a more practical (relative) scale on the sideo and/or rescale some graphs.
Minor:
- notations and graphical rendering of equations should be improved, and typographic unclear marks cleaned: see:
- eq. (4)
- "Ris" on line 98
- subscripts on line 76, 154, 162
- eq 21 and 22
- formula in line 178, 272 and 273
- interline space in 108-112
- reference 28 and 29 are the same spanning over two lines
Author Response
The authors present a fundamental study about the potential evidences of non-adiabatic coupling in single-molecules (SM) spectroscopy, by adopting a model hamiltonian that includes a coupling with an external bath and it is dynamically coupled to external radiating field. By exploring different regimes of coupling, the authors highlight when non-adiabatic coupling between electronic states manifests itself. The text is enough clear, and the physics of the presented results is sounding, although the impact on interpretation of experiments is only vaguely discussed and should be probably emphasized.
Answer. To our knowledge, electronic couplings leading to ultrafast electronic population dynamics have not yet been detected in femtosecond SM experiments. We argue that these couplings are manifested by an electronic population recurrence which can be detected by femtosecond SM spectroscopy in the strong-field regime. This phenomenon is so far a theoretical prediction which may turn out to be helpful for the interpretation of future femtosecond SM experiments.
The introduction of stocastic modulation of electronic parameters sounds a bit critical in the current form, since: (a) the modulation of each parameter is completely uncorrelated one from each others, (b) the amplitude of the stochastic modulation double the numbers of parameters, (c) the added physics is negligible (Fig. 2 and 3).
Answer. So far, we have performed simulations only with statistically uncorrelated modulations of the parameters. The generation of correlated modulations would require a microscopic molecular dynamics model of the polymeric environment, which is beyond the scope of the present work. On the other hand, as we argue in previous work [Ref 11 of the manuscript], useful information about (possible) correlations of the parameters could be obtained by recording a systematic series of SM signals. In general, the available SM data of the van Hulst group indicate that the parameter fluctuations are important, but the SM transients are not buried in noise. The amplitude of the noise chosen in the present simulations may be a bit low, but the effects discussed in the manuscript are nevertheless clearly visible.
About the parametrization and the choice of the model hamiltonian, the authors might find interesting some other work in the literature, quite relevant even if applied to a different phenomenon (see for instance Di Maiolo et al. [J. Chem. Theory Comput., 2018, 14 (10), pp 5339–5349], and other works by the same group).
Answer. We thank the reviewer. This paper is cited in the first paragraph of Sec. II of the revised manuscript with appropriate explanatory remarks.
I found somehow confusing the usage of different scale for intensity in Figures 2(a-d) and 3(a-d), making less easy to read the extent of the fluctuations and the comparisons. The authors could consider to add a more practical (relative) scale on the sideo and/or rescale some graphs.
Answer. In the weak-field limit, the signals scale linearly with the intensity of the pump pulses and therefore the signals can be rescaled without loss of information. In the strong-field regime, on the other hand, this is no longer the case, and relative intensities of the signals are relevant and have a physical meaning.
Minor:
- notations and graphical rendering of equations should be improved, and typographic unclear marks cleaned: see:
- eq. (4)
- "Ris" on line 98
- subscripts on line 76, 154, 162
- eq 21 and 22
- formula in line 178, 272 and 273
- interline space in 108-112
- reference 28 and 29 are the same spanning over two lines
Answer. We made the corrections as far as we could identify mistakes or typos in the text.